# Healthy eating and physical activity: Analysing Soweto's young adults' perspectives with an intersectionality lens

**Gudani Mukoma**[1,2]*, **Edna N. Bosire**[1,3], **Sonja Klingberg**[1], **Shane A. Norris**[1,4]

**1** Faculty of Health Sciences, Department of Paediatrics, SAMRC/Wits Developmental Pathways for Health Research Unit, University of the Witwatersrand, Johannesburg, South Africa, **2** Faculty of Health Sciences, Department of Biokinetics, Recreation and Sports Science, University of Venda, Thohoyandou, South Africa, **3** Brain and Mind Institute, Aga Khan University, Karachi, Pakistan, **4** Faculty of Medicine, School of Human Development and Health, University of Southampton, Southampton, United Kingdom

* gudani.mukoma@wits.ac.za

**Data Availability Statement:** The dataset used and/or analysed during the current study are attached as supporting information by the corresponding author for free access.

## Abstract

### Background and objectives

Non-communicable diseases (NCDs) are taking a toll on Africa's youth at younger ages than in other regions. These are attributed to risk factors that usually advance in adolescence, such as unhealthy diets and reduced physical activity. Young adults in South Africa, particularly women, tend to be sedentary, consume energy-dense diets low in micronutrients, and are more likely to develop NCDs much earlier in life than those in high-income countries. With an intersectionality perspective, this study explored young adults' barriers and solutions to addressing these risk factors in Soweto.

### Setting

Soweto, Johannesburg, South Africa, is one of the most well-known historically disadvantaged townships known for its established communities, and socioeconomic and cultural diversity. *Design*: A qualitative investigation utilising focus group discussions (FGDs) with a topic guide. FGDs were transcribed verbatim and thematically analysed using a combination of deductive and inductive approaches.

### Participants

15 Men and 15 women 18–24 years of age living in Soweto (n = 30). *Results*: South African young adults have a basic understanding of the significance of nutrition, exercise, and their ties to health. However, numerous barriers (like taste, affordability and crime) to such behaviours were reported, arising from the participants' personal, domestic, social, and local community levels. Young women experienced sexism and had safety concerns while exercising in the streets, while young men tended to describe themselves as lazy to engage in exercise as they find it boring.

**Funding:** This work was supported by the National Institute for Health Research (NIHR) (GHR: 16/137/34 to S.A.N) using UK aid from the UK Government to support global health research. GM is a scholarship recipient of the DSI-NRF Centre of Excellence (CoE) in Human Development. The funders had no role in study design, data collection and analysis, decision to publish, or preparation of the manuscript.

**Competing interests:** The authors have declared that no competing interests exist.

## Conclusions

Young adults face a multitude of intersecting barriers, making it difficult to adopt or sustain health-promoting behaviours. It is important that potential solutions focus on the intersections of barriers to healthy eating and physical activity in order to provide more realistic support for such behaviours.

## Introduction

Non-communicable diseases (NCDs) account for 17 million deaths annually, 86% of which are premature and take place in low- and middle-income countries [1]. NCDs like heart disease, stroke, and diabetes are becoming more prevalent causes of death and disability in sub-Saharan Africa (SSA). By 2035, NCDs are anticipated to surpass infectious diseases as the main cause of death and disability [2,3]. Particularly concerning is the fact that in Africa, NCDs are affecting people at a much younger age (10 or more years younger) than in wealthier regions [4]. This rising burden of NCDs is the result of risk factors that typically begin in adolescence, such as unhealthy diets, low physical activity (PA), and sedentary behaviours, which are driven and exacerbated by urbanisation, globalisation, and commercial health determinants [5–8]. Furthermore, research evidence suggests that the health risks are transferred to the next generation as well as affecting young people themselves in their adult age [9]. To prevent the development of NCDs in future generations, the 2016 Lancet Commission on Adolescent Health and Wellbeing recommended investing in prevalent NCD-related health behaviours among adolescents [10].

Young adults (18–26 years) in South Africa, particularly women, tend to lead more sedentary lives compared to their male counterparts, consume energy dense but micronutrient poor diets, and are at a high risk of developing NCDs much earlier in life [11–14]. Due to behavioural health risks, women were twice as likely as men to die from heart disease between 2012 and 2016 (14.7% vs. 7.1% respectively) [15]. A recent survey in urban South Africa found that 46.6% of young women aged 18–24 years were in the body mass index-based categories of overweight or obese, and 42.5% of them did not engage in regular physical activity [16]. On the other hand, consumption of foods (fats, oils, sauces, dressings, condiments, sweets and savoury snacks) associated with excessive weight gain has increased by 30% over the past decade [17]. While NCD risk factors are already high among young women, they significantly increase with age [18,19]. It is therefore important to support young people to adopt and maintain behaviours that minimise NCD risks, setting them on healthier trajectories. In order to do so, we must first gain an understanding of the contextual factors influencing their health beliefs, diet and physical activity behaviour.

Individual characteristics, as well as, social and physical environment factors such as home, family, peers, neighbourhood, school, and workplace, have been shown in studies to influence health beliefs and behaviours [20–22]. From previous studies in urban South Africa, we learned that perceived affordability, taste, availability, individual preferences and family were significant determinants of food choices [21,23,24], while physical activity opportunities are constrained by a lack of facilities and safety concerns [21]. These barriers have been shown to vary by age and gender among children and adolescents from high-income countries [25]; however, evidence of how they differ among young adults is scarce. Changes accompanying the transition from adolescence to adulthood may make individuals more susceptible to emotional and social influences, making them more likely to favour behavioural choices that will

benefit them now rather than in later life [13,16,26,27]. When this is compounded by the complexity of overcoming a multitude of barriers that arise from the social and physical environment, engaging in healthy behaviours becomes much more difficult [28]. In contrast to other studies that have used socio-ecological models to demonstrate the different levels of influence on health-related behaviours, we used an intersectional lens in order to explore how young people face multiple connected barriers to healthy behaviours [29–32]. The aim of this study was therefore to explore young adults' barriers to healthy eating, physical activity, and the solutions to address these barriers with an intersectionality perspective. The intersectionality perspective will be useful in clarifying how personal, domestic, social, and environmental barriers are interconnected and occur concurrently to disadvantage young people in making healthy behaviour choices.

## Methods

### Setting and study design

This cross-sectional qualitative study was conducted at the South African Medical Research Council (SAMRC)/Wits Developmental Pathways for Health Research Unit (DPHRU) at the Chris Hani Baragwanath Academic Hospital (CHBAH) in Soweto. CHBAH is the largest hospital in Africa and the third worldwide; it is also a public tertiary care institution that serves the low-income community of greater Soweto in south-western Johannesburg, South Africa [21]. Soweto is one of the most well-known historically disadvantaged townships in South Africa known for its established communities, and socioeconomic and cultural diversity [33]. However, while there is economic diversity in Soweto, poverty-related challenges are still a reality, including unemployment and food insecurity [34], and poor access to appropriate health services, especially for young adults [35].

Focus group discussions (FGDs) were utilised to collect data. We used a questionnaire to record the demographic information and anthropometric measures of participants. Participants were measured while wearing loose clothing and without shoes for all anthropometric measurements. A Seca 213 portable stadiometer was used to measure the participants' height (in cm), which was then translated to meters (m). Using a portable electronic bathroom scale, body weight was calculated to the nearest 0.1 kg. Weight in kilograms (kg) divided by height in meters ($m^2$) was used to determine body mass index (BMI). These measurements were taken during the quantitative survey before the focus groups began.

### Participants and recruitment

We purposively recruited and enrolled thirty apparently healthy black men and women between the ages of 18 and 24 years from Helti's Soweto household enumeration study database [36], and also from Soweto households that had never been pregnant or fathered a child. The Soweto household enumeration database provided a sampling frame, which enabled identifying participants from both low and high-income households hence ensuring participants' socioeconomic diversity [36,37]. Socioeconomic status (SES) was determined for those newly recruited from Soweto households using a household asset index described in detail elsewhere [16]. Participants who have previously conceived and fathered a child (including miscarriage and stillbirth) were ineligible to take part. During recruitment, research assistants contacted prospective participants using the contact information provided on the enumeration study database and conducted household visits in Soweto communities inviting eligible participants to participate. They outlined the purpose of the study, as well as provided contact information in case more information was required. Research assistants at DPHRU collated the information on prospective participants who were then called back to make appointments. The

recruitment of these participants was to form part of a larger package of work by the Global Diet and Activity Research (GDAR) Network which started in the year 2020, explained in more detail elsewhere [37,38].

## Theoretical framework

The theoretical framework for this study is provided by an understanding of the Socio-Ecological model of health and the Theoretical Domains Framework (TDF) with an intersectionality perspective, which shows that health-related behaviours are influenced by physical, biological and psychological, sociological, economic and cultural barriers [39–42]. The foundation of this paper is the assertion that various barriers intersect to affect how health-related behaviours turn out, which is supported by both the social-ecological model of health and the TDF [39–42]. We applied the intersectionality framework to comprehend how context influences the emergence of health problems in complex ways rather than reducing perceptions and behaviours to a single characteristic [40]. To be more precise, we explored different factors (personal, domestic, societal, and neighbourhood) that affect people's decisions and beliefs and induce particular behaviours such as unhealthy eating and physical inactivity [39,40,42]. The intersecting rings of the model (Fig 1) illustrate how factors at one level intersect with factors at another level and influence behavioural outcomes. For instance, easy access to unhealthy food may intersect with a person's taste preference, leading to higher consumption of unhealthy foods. In recent years, there has been a strong push to develop environmental level health interventions such as school-based physical activity promotion to expand beyond programs that exclusively target individual level behaviours [43]. Whilst research has extensively reported on the importance of households, parental feeding practices and family influence in shaping childhood health habits [44], the same cannot be said about adolescents and young adults [45]. As young adults transition to adulthood, the personal, social, and environmental changes that transpire such as moving out of their home, studying, cohabitation with peers or partners, and starting a new job have a great influence on their lifestyle behaviour choices that are linked with poor diets and rapid weight gain [46]. Therefore, interventions that target the environment may be more efficient and potentially more effective than individually targeted interventions because they are designed to change the context in which people live and work to create conditions supportive of healthy behavioural choices [41]. In addition, although working with individuals to affect behaviour is difficult and resource consuming, interventions that influence policies and group-level behaviours can in turn affect individual-level behaviours among a much larger group of people and thus be more resource efficient [41,47]. Thus, health interventions targeting young men and women should take into account all intersecting elements [33].

## Data collection

The data were collected during May and June of 2020. The FGDs were conducted in English by two experienced multilingual qualitative researchers, one facilitator and one note taker, with participants using vernacular where necessary. All FGDs were held remotely using Zoom Meetings (an online video conferencing platform) due to the COVID-19 pandemic and the necessity to ensure the safety of our participants. A day before the FGDs, all participants received a one-hundred-rand ($5,50) airtime recharge voucher to cover internet costs. Across all four FGDs, a topic guide (S1 File) was used to elicit discussions around perceived barriers and solutions to healthy eating and physical activity. A total of four FGDs were conducted, each comprised of 7–8 participants, and lasted between 45–90 minutes. In order to maximise disclosure among focus group participants, the groups (G) were divided by gender, with two

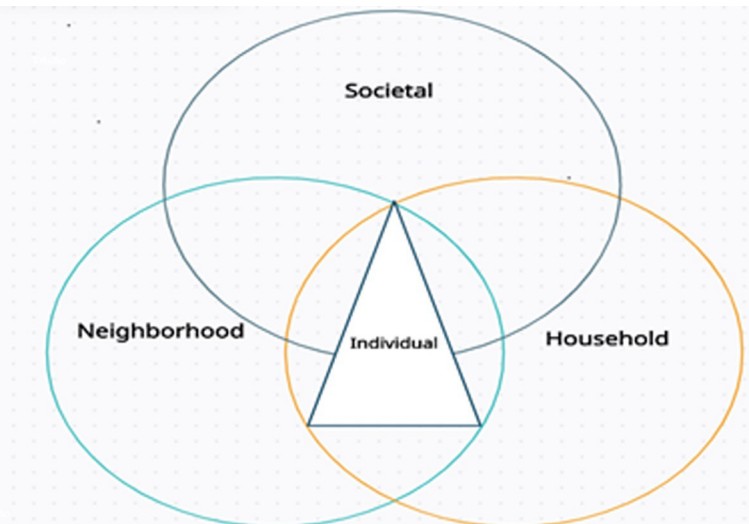

**Fig 1. A conceptual diagram developed for this study showing different factors intersecting to influence young individuals health related behaviour.**

groups made up of young men and two groups made up of young women (all aged 18–24 years). All FGDs were audio-recoded with participants' permission to record. The FGD sessions were meticulously documented, including observations. Audio recordings were transcribed verbatim and, when necessary (i.e., views shared in native languages), translated into English. All participants provided an electronically signed informed consent before taking part in the study. The Human Research Ethics Committee (HREC) at Witwatersrand University granted ethical clearance, with ethics number M190523.

## Data analysis

Data were analysed through thematic analysis [48,49] with the help of the qualitative analysis software MAXQDA 2020 (version 20.4.2). Initially, data familiarisation was done by the first author, which involved checking transcripts and repeated reading of transcripts. Thereafter, coding, initial themes, reviewing themes, defining and naming themes (S2 File), and finalising the analysis followed [50]. Other authors were asked to review the initial themes; the reviews were used to refine the key themes until an agreement was reached. The data were coded and analysed using a combination of deductive and inductive approaches. The deductive approach was based on pre-identified themes that focused on the research question, while the inductive approach was used for all themes generated from the transcripts and field notes [49]. This process enabled an interpretation of young adults' perceptions of various barriers to healthy eating and physical activity, and specific solutions to address these barriers. The key identified domains in relation to barriers and solutions to healthy eating and physical activity are described in the results section.

## Results

### Sample characteristics

The analysis included a total of 30 participants (15 men and 15 women). The average age of the men was 21.5 years, that of the women 20.5 years. Compared to the young women (46.7%), more than half of the young men (53.3%) had completed grade 12 or high school. Overall, only

3.3% of the participants had a university degree. There were significant differences in prevalence of overweight and obesity which was higher among women (53.3%) as compared to the men (0%) (Table 1).

We present our findings from thematic analysis under the following four domains and nine main themes (in brackets): (1) Individual level factors (comprehension of health concepts and their interrelations; inability to apply nutrition knowledge to action; personal circumstances influence eating habits; not motivated enough to be active); (2) Influence at the household level (priority is for everyone to eat; immersed in household chores, no time for exercise); (3) Social influences (eating healthy means sickness) and (4) Influences by neighbourhood environment (junk food is inescapable, it's everywhere and cheap; no gyms, exercising outside is unsafe). Additional extracts of participants' views can be found in (Tables 2 and 3).

## Domain 1. Individual level factors

**Comprehension of health concepts and their interrelations.** In the various group discussions, young adults displayed a good understanding of health concepts and their interrelationships. They described health as a daily process involving routine check-ups, sufficient sleep, a healthy and balanced diet, and physical activity to promote or maintain overall wellbeing. *"Health is how you take care of your body everyday like seeing the doctor to check BP [blood pressure] and with the kind of food that you eat like veggies and gym and the time you sleep. It is the wellbeing of your whole body." (G2 men).*

When talking about nutrition, many young adults indicated that nutrition is about the foods people eat, which can be good or unhealthy. This in their opinion meant that nourishment could be both good and ill. Following a specific dietary plan that included eating fruit and foods high in protein and vitamins was generally recognised as good nutrition; *"[Nutrition is] a diet with proteins and vitamins that a person should follow like eating fruit and eating vegetables to stay healthy." (G3 women).* Good nutrition is important for healthy growth, a robust immune system and a "sharp mind" [cleverness and quick thinking] according to participants' views; *"Food that helps our body in terms of growing strong and healthy, and also having a strong immune system and sharp mind". (G1 women).* On the contrary, eating a modest amount of unhealthy food (referred to as "junk") was not considered a health risk by some young people, but eating too much of it was: *"I believe it is about how you eat the junk that is bad. If you have too much of junk food your health is going to be bad. But if watch what you eat then it's not bad." (G1 women).*

**Table 1. Characteristics of young adult focus groups discussion participants in Soweto.**

| Variables | Total (n = 30) | Men (n = 15) | Women (n = 15) |
|---|---|---|---|
| Mean age | 20.9 (2.1) | 21.5 (2.1) | 20.5 (1.9) |
| **Education level** | | | |
| Secondary schooling | 13 (43.3) | 5 (33.3) | 8 (53.3) |
| Passed grade 12 | 16 (53.3) | 9 (60) | 7 (46.7) |
| Tertiary | 1 (3.3) | 1 (6.7) | 0 |
| **BMI Categories** | | | |
| Underweight | 1 (3.3) | 1 (6.7) | 0 |
| Normal weight | 21 (70) | 14 (93.3) | 7 (46.7) |
| Overweight | 5 (16.7) | 0 | 5 (33.3) |
| Obese | 3 (10) | 0 | 3 (20) |

Values are presented as mean and standard deviation (SD), frequencies and percentiles (%).

**Table 2. Young adults' understanding of health concepts.**

| Domain | Theme | Excerpts from women | Excerpts from men |
|---|---|---|---|
| Individual level factors | *Comprehension of health concepts and their interrelations* | *"Health simply means taking care of your inner person like your body by eating healthy and exercising making sure you avoid stress."* (G3 women) | *"Health is how you take care of your body everyday like seeing the doctor to check BP and with the kind of food that you eat like veggies and gym and the time you sleep. It is the wellbeing of your whole body."* (G2 men). |
| | [nutrition is] | *"A diet with proteins and vitamins that a person should follow like eating fruit and eating vegetables to stay healthy."* (G3 women) | *"Eating fruit, mostly planted fruits and healthy food sort of like proteins and veggies."* (G2 men) |
| | | *"Food that helps our body in terms of growing strong and healthy and also having a strong immune system and sharp mind".* (G1 women) | *"I believe it is about how you eat the junk that is bad. If you have too much of junk food your health is going to be bad. But if watch what you eat then it's not bad."* (G1 men). |
| | [Physical activity is] | *"Physical activity is part of our day-to-day life. It can be when you wake up in the morning take a walk or jog, play soccer or go to gym."* (G2 men). | *"Physical activity is part of our day to day life. It can be when you wake up in the morning take a walk, do cleaning, play soccer or go to gym."* (G2 men) |
| | [Obesity is] | *"A drastic amount of fat in a human body. It can actually kill you as it brings heart problems and high BP."* (G3 women) | *"Obesity is a big problem because now you eat too much junk and you just get fat and very sick"* (G2 men) |

In the discussions about physical activity, the young adults mostly defined it as participation in daily planned and organised exercise routines or sporting activity; *"Physical activity is part of our day-to-day life. It can be when you wake up in the morning take a walk or jog, play soccer or go to gym."* (G2 men). Some of the reported benefits of physical activity included strengthening muscles, maintaining good posture and preventing disease, especially heart disease; *"Physically activity, we also need that because that strengthens the muscles which ensures healthy posture and prevents diseases."* (G3 women). However, the young adults did not see unstructured and unplanned tasks such as cleaning, cooking or laundry as physical activities, but rather as barriers, that takes up a large part of their daytime and leaves them no time for exercise;*"I'm always busy like cooking and cleaning you know house duties and get tired so where do I get time to exercise."* (G3 women).

In many cases, young adults mentioned that physical inactivity was associated with obesity burden and increased risk of NCDs. *"[high obesity risk is] when you are a couch potato [always sited] and don't exercise you gain fat. . . . . . you easily get sick with BP [high blood pressure] and heart problem when fat."* (G4 men). Physical activity was recognised as the only option for maintaining good health. From participants' narratives, physical activity is a more effective means of controlling obesity and preventing diseases than diet; *"Even if you eat junk foods, you can be healthy and safe from obesity and diseases if you get exercise."* (G4 men). In several instances, participants also acknowledged that genetics is a factor contributing to the burden of obesity *"Sometimes the obesity is taken from other people in your family."* (G4 men) demonstrating a knowledge of risk factors outside their control.

Further, in the discussions, participants generally agreed that obesity meant a large amount of fat in the body that causes NCDs and could lead to death: *"[Obesity is] A drastic amount of fat in a human body. It can actually kill you as it brings heart problems and high BP."* (G3 women). Undoubtedly, young adults recognised obesity as a significant burden in their communities, but they believed that obesity had been normalised, as even those affected perceived it as a signal of contentment rather than a health risk. *"Yes, obesity is a problem where I live, but many people that are fat are fine with that and say it shows they don't have problems [social and financial]."* (G3 women).

**Inability to apply nutrition knowledge in practice.** Across all FGDs, the majority of participants expressed that they did not know how to translate their knowledge about nutrition

**Table 3. Young adults' perceived barriers to healthy eating, physical activities and their solutions.**

| Domains | Perceived barrier | | Solution to perceived barrier | |
| --- | --- | --- | --- | --- |
| | Theme | Extracts | Proposed solution | Extracts |
| **Individual level factors** | *Inability to apply nutrition knowledge to action* | "The thing that we lack most is the knowledge of how healthy foods are important to our body and our life." (G2 men) | Provide education programs on healthy eating in local languages | "We can start TV or radio programs educating what is obesity in our language and what are the effects of obesity instead of just saying we need to eat healthy." (G1 women) "To change that thing we need to have programs even in schools where people will be taught the importance of eating healthy foods."(G2 men) |
| | *Personal circumstances influence eating habits* | "When they cook healthy stuff, we don't eat the veggies because they taste bad." (G2 men) "The healthy food taste weird, not nice." (G3 women) | Teach them young about healthy living | "Teach them at a very young age to eat healthy and to go to gym exercise more. Because if you don't teach them at an early age they won't get used to it." (G2 men) |
| | | "We eat Kota; you don't need to wait for them to cook especially when it's cold days." (G1 women) "Come on guys' fruit is money, veggies you also need more money but Kotas are cheap." (G1 women) | Government subsiding healthy food to make their access easy | "Government must subsidise healthy food so that it can be affordable to those who want to eat healthy food but can't afford." (G3 women) |
| | | "If at home they don't teach you how to eat healthy you will never be healthy because you are going to get used to Kotas." (G2 men) | Eating together as a family | "We all eat together in my family; I am forced to eat the greens I mean veggies like cabbage, spinach and potato. But alone, I love to eat Kota." (G2 men) |
| | *Not motivated enough to be active* | "I do know about exercising just that I'm lazy and just don't want to exercise, it is boring." (G2 men) "We have facilities to exercise in my community, for me is ignorance and laziness that is all." (G3 men) | * Partner with friends or family * Forming community clubs | "I sometimes exercise with my friend in the morning and with my two older brothers in evenings. They make it fun and safe to exercise I never miss." (G1 women) "I think it is much better if it is done in groups like start a community club to exercise same with soccer and netball." (G1 women) |
| | | "In your back yard you are required to do the exercise alone and you are discouraged." (G3 women) | Physical activity coaching | "We can have meetings with coaches like 3 days per week where there comes coaches to motivate and they teach you to be physical activity and show some exercise." (G2 men) |
| **Influence at the household level** | *Priority is for everyone to eat* | "From childhood we grew up in an home that is not knowledgeable, we are just living our lives of eating junk food like Kotas." (G2 men) | Being independent linked to freedom of food choice | "Start taking responsibility of your life, when you start having your own place, you start buying groceries for yourself." (G4 men) |
| | | "I am the only one that wants to eat those healthy foods at home." (G2 men) | Teaching healthy eating at home | "We should teach the young ones at home first about how do we eat healthy so they won't fall in the trap of eating junk food." (G2 men) |
| | | "At home you can't tell them what to eat and what not to eat you see. You eat what is there" (G1 women) | Financial support linked to choice of diet | "I feel like is when I'm given my own money at home I can buy green food and fruits." (G1 women) |
| | | "The background where we live is poor, I think we like to live healthy but they can't afford to buy healthy food." (G1 women) | Community feeding schemes | "I think if they at the moment, we need to have like a community feeding scheme, like food parcel with veggies and fruits." (G2 men) |
| | *Immersed in household chores, no time for exercise* | "Sometimes it is not easy for me if I can just have two minutes of my time to exercise in my household." (G1 women) "I'm always busy Like cooking and cleaning you know house duties and get tired so I don't get time to exercise." (G3 women) | Become independent | "Start taking responsibility of your life, when you start having your own place, you start buying groceries for yourself." (G4 men) "Staying alone is better, less duties and more time for exercising." (G3 women) |

*(Continued)*

**Table 3.** (Continued)

| Domains | Perceived barrier | | Solution to perceived barrier | |
|---|---|---|---|---|
| | Theme | Extracts | Proposed solution | Extracts |
| Social influence | *Eating healthy means sickness* | *"The majority of them [peers/community members] believe if you eat healthy stuff like veggies right, it means that you are sick with things like with HIV/AIDS or something you know so I rather not eat." (G4 men)* | Start nutrition group discussions | *"Make a group discussion with people and educate each other on nutrition." (G3 men)* |
| Influence by neighbourhood environment | *Junk food is all over and cheap* | *"A lot of places all you see is junk foods, every street they selling bunny chows and fat cakes." (G3 women)* *"There are pictures of KFC all over, taxi ranks and by the road to Maponya [local shopping mall] too you just want it." (G1 women)* | * Starting community gardens * Promote healthy eating through adverts | *"In our community we do gardening, So we can provide those veg and fruit to those in need within our community." (G3 women)* *[We need] "Adverts to promote more healthy living tips or healthy foods and less unhealthy foods like more veg not beer." (G4 men)* |
| | | *"It is difficult for me to eat healthy you find that the shops in my area are selling all the junk foods." (G1 women)* *"Around me there is a Spaza that sells tempting food like the Kota." (G1 women)* | * Healthcare professionals providing awareness on junk foods * Treat unhealthy eating like a pandemic | *"Health workers can go to fast food places like Spazas and then teach people instead of having all the junk foods rather go this way to help your health." (G1 women)* *"The government must address the nation like with Covid, address with a couple of food that is unhealthy and show healthy one, do same like hand sanitizers when you go to the shop you know it is law." (G3 women)* |
| | | *"In my area I can't jog outside, there are always boys sitting at the corner saying embarrassing things about my body and their presence makes me feel uncomfortable." (G1 women)* | Government building infrastructure for physical activity | *"Government must hire security officers in community parks so like training equipment are not broken and stolen by Nyaope [A local street drug, created by combining heroin, cannabis and other substances] boys. We just want to feel safe you know." (G3 women)* |
| | *No gyms! Exercising outside is unsafe* | *"I believe like to be in a physical activity I need to go to gym and things like that which is something that is not there in our area." (G1 women)* | Use space around you | *"Use the parks to gym, you can play soccer, you can use the streets, you can run. And you can actually just gym inside your yard." (G2 male)* |
| | | *"I want to wake up at 6 and jog but if you are a girl you can't do it at night, it is not safe." (G1 YA women)* *"In my community there was this one issue where one woman, when they were busy training or exercising at the park they were raped." (G3 women)* | * Security officers in parks * Community patrol forum for protection | *"Government must hire security officers in community parks so like training equipment are not broken or stolen." (G2 men)* *"The only solution—have the community patrol forum looking after us during that time of physical exercises that we will be having." (G4 men)* |

and physical activity into healthy behaviour; *"The thing is we don't know a way to use this knowledge of how healthy foods are good to our body, at the same time our places [home and neighbourhood] make it hard to do exercise." (G2 men)*. One proposed solution to address this barrier was to provide health education in local languages, covering topics such as nutrition and physical activity through community TV and radio stations, social media platforms (WhatsApp, Facebook and Twitter) and integrated into school lessons, especially at the primary level (grades R-3). This solution, in the participants' opinion, is beneficial as it targets more than one barrier and takes into account language, outreach and accessibility issues that they have already encountered in other educational programmes:

*"We can start using TV or radio programs in the community to teach about healthy food and physical activity. Also periods in primary classes like grade R or all lower classes can also teach*

*what is obesity in our language and what are the effects of obesity and what to do to avoid it instead of just saying we need to eat healthy and exercise." (G4 men).*

**Personal circumstances influence eating habits.**   From participant discussions, it appeared that their basic understanding of health concepts, especially nutrition and physical activity, did not translate or influence an intent to engage in healthy behaviours. In all FGDs, participants admitted that their personal preferences, such as the taste of healthy foods, addiction to junk food, price and convenience of unhealthy foods were the main drivers to unhealthy eating patterns. In other cases, young people mentioned that they were able to eat both healthy and unhealthy foods, depending on circumstances such as affordability, taste and weather. For example, some reported that certain foods, such as those high in fibre, were of great benefit in cold weather. *"I do eat fruit and veg. I also eat Kota [A quarter loaf of bread stuffed with foods such as fried chips, pork sausage, cheese, polony, tomato sauce and mango atchaar] when it's cold days [because it is] high in fibre to keep warm." (G1 women).* To overcome these barriers, participants suggested practising and promoting family meal sharing, as they recognised that time as an excellent opportunity for parents or adults in the family to teach the young one's about healthy eating. *"We all eat together in my family, we eat the same food. I am forced to eat the greens I mean veggies like cabbage and spinach, [that] mom cooks a lot." (G2 men).*

**Not motivated enough to be active.**   Further in the discussions, participants cited personal feelings such as laziness, discomfort and the monotony of exercise as the main reasons that prevent them from engaging in physical activity. *"I do know about exercising just that I just don't want to exercise, it is boring and the pain, no." (G2 men).* Some participants expressed interests in PA engagement; however, a lack of motivation was a barrier to their participation. *"In your back yard you are required to do the exercise alone and you are discouraged." (G3 women).* Getting external motivators was proposed as a solution that could influence physical activity participation. According to participants, exercising with family or friends, or even joining a community club, might provide good motivation and promote regular participation in physical activity. *"I sometimes exercise with my friend in the morning and with my two older brothers in evenings. They make it fun and safe to exercise, I never miss." (G1 women).*

## Domain 2. Influence at the household level

**Only mom works, priority is for everyone to eat.**   Some young adults' expressed a desire to lead a healthy lifestyle, but family and domestic barriers such as the inability to make decisions on what food to eat at home, limited food options due to household size, poverty and lack of family support to eat healthily were the main barriers that always put a halt to such a thought; *"The household where I live is poor, only mom works, I think I like to live healthy but they can't afford to buy healthy foods just for me alone." (G1 women).* At this point, they declared that there was nothing else to do; moving away from home to become independent was the only solution to overcome these barriers. For them, independence meant taking control of their lives and deciding for themselves what they consume; *"Like when you start having your own place, you start buying groceries for yourself." (G4 men).*

While being independent was one of the proposed solution to unhealthy eating, discussants argued that young children who are heavily dependent on their parents or relatives are a greater concern as they cannot make this choice of moving out of the family. They commented on how providing unhealthy foods to young children with the notion that it is a good way to pamper them increased their likelihood of health issues. Participants noted that many young

children grow up believing that eating unhealthy foods is a form of self-indulgence because they have learned it at home; *"Small children are growing with obesity because of the junk like KFC they [caregivers/parents] provide to them, they believe in spoiling [a nice treat] them with that KFC." (G1 women).* To overcome this barrier, the young adults indicated that practising and teaching healthy eating at home is a much more viable strategy that can reduce the risk of premature mortality and diseases caused by eating poor quality meals from an early age. *"We should teach the young ones at home first about how do we eat healthy so they won't fall in the trap of eating junk food that causes diseases that can kill them later." (G2 men).*

**Immersed in household chores, no time for exercise.**   Discussions around physical activities in the context of other everyday activities at home were also extensively noted. Our findings show that men and women face different challenges in regard to physical activities within the home. In all FGDs, young adult women reported that they found it difficult to find time to exercise because of other household responsibilities such as washing or cleaning, helping to cook or caregiving roles. "I'm always busy like cooking and cleaning you know house duties and get tired." (G3 women). On the other hand, participants noted that men who could not exercise were just lazy, as they were not expected to do most of the household chores. *"I do know about exercising just that I'm lazy and just don't want to exercise, it is boring." (G2 men).* Becoming independent was their proposed solution to overcome this barrier because staying alone meant less chores. *"Staying alone is better, less duties and more time for exercising." (G3 women).*

## Domain 3. Social influence

**Eating healthy means sickness.**   Participants expressed their opinions on how peers, community members, and people in their communities have a general negative perception that eating healthy food is a sign of sickness. In particular, eating vegetables and losing weight are seen as indicators of illnesses like HIV/AIDS or those struggling with NCDs like diabetes and hypertension. As such, gaining weight and consuming unhealthy foods like "Kotas" seemed ideal because it meant no judgment from others; *"You know you start to lose fat when you eat veggies right. The majority of them [peers/community members] believe if you eat healthy stuff like veggies right, it means that you are sick with things like AIDS, so I rather not eat." (G4 men).* This was another way on in which consumption of unhealthy food was normalised. Participants proposed that nutrition education that can be delivered through group discussions in community can be a solution to create awareness and address this barrier. Accordingly, the discussions would encourage learning from one another, improving their understanding of the concept of healthy eating and associated benefits; *"[The solution is]. . ..make a group discussion with people and educate each other on nutrition and the benefits so they don't judge you." (G3 women)*

## Domain 4. Influence by neighbourhood environment

**Junk food is all over and cheap.**   Discussants reported that most street vendors and tuck shop sellers primarily sell unhealthy foods (commonly referred to as "junk") at lower prices. In addition, they noted that the high visibility of advertisements for unhealthy foods on main roads and neighbourhood streets exacerbates unhealthy eating habits. Based on their accounts, such visualizations, advertisements and proximity to junk foods have influenced unhealthy eating especially amongst young people:

*"When it comes to changing the situation and eat healthy you can't because of the neighbourhood that you live around, we are exposed to junk food all over and it's cheap at the spaza*

*[neighbourhood convenient tuck shop]. As you walk down the road all you see is a big bill-board with coke, burger and chips so you always think of eating them." (G2 men).*

These challenges led participants to propose setting up community gardens as a solution. They also mentioned a need to have more healthier food advertisements and awareness campaigns by health professionals about the dangers of unhealthy foods. Others argued that unhealthy eating should be considered as a pandemic, meaning government regulations on unhealthy food must be imposed in order to address these neighbourhood-level barriers; *"The government must address the nation like with Covid, address with a couple of food that is unhealthy and show healthy one, do same like hand sanitizers when you go to the shop, you know it is law." (G3 women).*

**No gyms! Exercising outside is unsafe.** While discussing about the environmental barriers to physical activity, it was noted that structural barriers, such as a lack of access to gyms or facilities for physical activity were among the most prevalent challenges that young adults' experienced; *"The fact is that we don't have places that we can go maybe to exercise after eating that Kota". (G4 men).* Participants' narratives also revealed gender disparities, with women generally fearing for their safety and encountering sexist remarks while exercising around the neighbourhood. *"In my area I can't even jog outside, there are always boys sitting at the corner saying embarrassing [sexist remarks] things about my body and their presence makes me feel uncomfortable, I don't feel safe." (G1 women).* To address these barriers, participants proposed that young women should be encouraged to use their household backyards for physical activity. In addition, participants noted a need for the government to construct more facilities for physical activity in the community as well as employing security officers in community parks, or having a community patrol to man security. This was said to be beneficial to all community members and may influence many young people to engage in physical activity.

## Discussion

We set out to explore young adults' barriers to healthy eating, physical activity, and the solutions to address these barriers with an intersectionality perspective. Findings from the FGDs suggested that young adults in this urban South African township understood the significance of a healthy diet and physical activity and their relationship with health. However, they felt unable to put their knowledge into practice. Young adults in this study reported several barriers to healthy eating including taste and preference; a lack of autonomy on food decisions at home; negative beliefs about healthy foods; street vendors, advertisements and convenience stores that primarily sell unhealthy foods at lower prices. The commonly discussed barriers to physical activity were laziness; lack of facilities; and safety concerns when engaging in PA in their neighbourhoods.

Furthermore, we found that barriers to physical activity differed by gender. On one hand, young women found it more difficult to exercise because of housework chores, experienced sexist remarks or they had safety concerns. On the other hand, laziness, discomfort and the monotony of exercise and lack of exercise facilities demotivated young men to engage in physical activity. That said, young adults proposed a number of solutions to addressing these barriers and among these were: the practice and promoting family meal sharing; providing health education in local languages; becoming independent; getting external motivators; practising and teaching healthy eating at home by adults; creating community gardens; more advertising of healthier foods; conducting more education campaigns by health professionals on the dangers of unhealthy foods; establishing a community patrol forum; and building accessible physical activity infrastructure by the government.

Our findings are consistent with those reported in studies conducted with adolescents, young women, and adults [19,21,22,41–44]. According to Ware et al., [22] in a South African study of urban young women, those who lived with their parents believed they had no influence over what was cooked and consumed in the home, and they perceived unhealthy foods to be cheaper, more accessible, and convenient because they were readily available. Among adults, Bosire et al. reported that the stigma associated with being thin, which links it to conditions such as HIV/AIDS, has a negative impact on people's risk perceptions of being fat or overweight and makes them want to gain weight in an effort to escape this stigma. Wrottesley et al. [19] also reported on how adolescent girls were discouraged from exercising because they were concerned about losing weight or becoming too thin. Our findings, as well as those of previous studies, support our hypothesis that the young adults in this study are confronted with multiple barriers arising from the macro and micro levels of society [41] (individual, family, community-level, grassroots institutions, and policies) that intersect to complicate their ability to engage in healthy behaviours, as shown in the conceptual diagram below (Fig 2), which illustrates how individual-level factors intersect with household-, neighbourhood-, and societal-level factors to influence behaviour. There is a need to understand the complex nature of these health inequities especially among young adults. Intersectionality is becoming more widely recognised as an important theoretical approach for studying health-related inequalities by highlighting the intersections of individuals' multiple identities within social power systems that exacerbate and reinforce experiences of poor health behaviours [42]. To the authors' knowledge, this is the first study carried out in South Africa that analyses the barriers that hinder young adults from engaging in physical activity and eating healthily as well as the solutions to overcome these barriers using an intersectionality framework.

Since the 1970s, neoliberal ideologies have grown in popularity and provide little support for addressing social determinants of health and wellbeing [51,52]. These ideologies emphasize individual attributes and choice as major drivers of health [53] and have led to a health promotion practice where interventions are implemented without taking contextual factors into account [49]. Intersectionality goes beyond looking at individual factors like biology, socioeconomic status, sex, gender, and race. It focuses on how these factors relate to one another and interact with one another at various societal levels to understand how health is shaped across

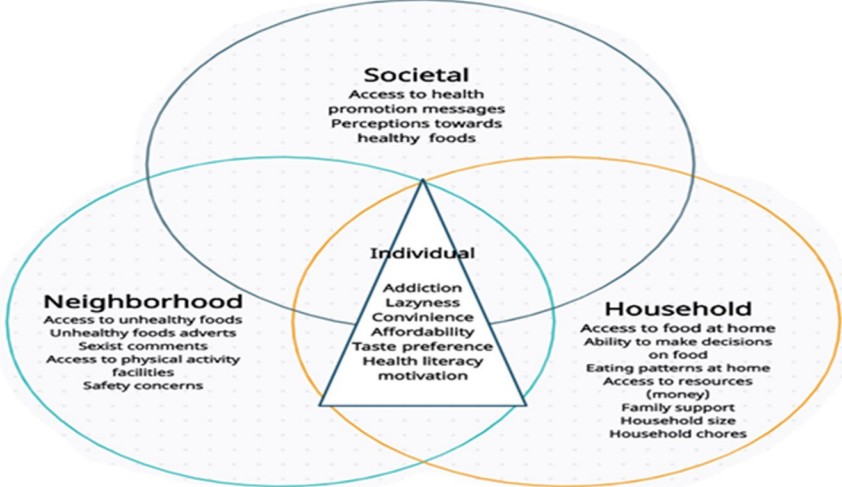

**Fig 2. Conceptual diagram showing the intersectionality between different multilevel barriers to healthy eating and physical activity influencing young adults' health related behaviour.**

demographic groups and geographical contexts [54]. In this study, intersectionality is employed to explain how individual, household, societal, and neighbourhood barriers to healthy eating and physical activity participation interact, as well as their proposed solutions. As Louis Althusser pointed out, people (in this case, young adults) are conditioned to not only hear messages but also obey, comply, and live up to them [55]. It is these context-based generated messages that address social ideas of race, class, gender, sexual orientation, age, and even physical characteristics often associated with disabilities or abilities in young adults [55]. This means that healthy eating and physical activity perceptions are more strongly influenced by the messages and meanings transmitted by meaning providers through social conditioning. Parents, friends, family and community members are among the people who can provide meaning, as do groups like those involved in school sports, churches or other religious institutions, and the media (visual and audio). Consequently, what young adults encounter in their families (lack of family support to eat healthily), neighbourhood or communities (easy access to unhealthy foods), and through media like as TV and music (advertisements of unhealthy foods) influence what they perceive as important and what they ultimately prioritise in a given society. At this point, intersectionality is when all of these categories cross paths, revealing how inequities are shaped by interactions between different sites and spheres of power, which allows us to observe how these are all interconnected on a larger scale. Considering the intersectionality of social constructs, identities, and histories, we can see that there is even more connection between barriers to healthy eating and physical activity at the individual, household, social, and neighbourhood levels. To fully understand one, we need to understand the others.

Furthermore, applying an intersectionality lens in this study contributes to highlighting significant differences between young men and women that are often overlooked when addressing barriers to healthy eating and physical activity. In particular, it offers an explanation to how young men and women from lower socioeconomic groups might decide to exercise outside because they lack access to gyms, but only the young women would encounter sexism and concerns about safety. Because of this, young men have a slight gender advantage over young women when choosing to exercise on neighbourhood streets. Furthermore, this may also explain why some health education interventions are unsuccessful despite being recognised by research as a crucial component in the management and prevention of NCDs and their associated risk factors [49]. For example, South Africa is a country with a wide range of linguistic and cultural diversity, yet health systems and public health interventions frequently use English as the main language for imparting health education, either orally or in handouts [51]. This is despite the evidence that shows there is a lack of health education among South Africans [14,50,51]. In this case, it becomes apparent that English-speaking people from lower socioeconomic groups may have a greater advantage in accessing health services and comprehend how health education pertaining to nutrition and physical activity can be applied than Africans who do not speak and understand English but have similar backgrounds. All this various aspects at different contextual levels interact and mutually inform one another and should be taken into account when implementing interventions or solutions that aim to influence healthy behaviour choices.

There are some limitations on the current study. The FGDs were held in the midst of the COVID-19 pandemic using online video conferencing software rather than where participants lived and went about their daily lives, so we were unable to supplement the data obtained through the FGDs with additional sources of information like field observations. Some conversations had to be halted due to unstable network conditions. The young men and women who formed part of the discussions did not practice healthy eating or engage in regular physical activity; however, holding discussions with those who have adopted health related behaviours might have been of interest.

## Conclusion

This study is important because it highlights the barriers to healthy eating and physical activity among young adults in Soweto and the proposed solutions. While some solutions have been proposed, young adults believe that they are not able to put them into practice. Our findings have also shown that young adults face a multitude of barriers that occur simultaneously at the individual, household, neighbourhood and societal levels, making it difficult for them to make healthy behavioural choices. We advocate for young people to be involved in developing or designing solutions to promote health or influence healthy lifestyles. By focusing on the intersection or overlap of different factors, findings of this study could be used to influence health-related programmes in Soweto and other similar contexts.

## Supporting information

**S1 File. Focus group discussion guide.**
(DOCX)

**S2 File Dataset.**
(MX22)

## Acknowledgments

The authors thank the participants who took part in this research and the Soweto GDAR research team whose work they represent here.

## Author Contributions

**Conceptualization:** Gudani Mukoma, Shane A. Norris.

**Data curation:** Gudani Mukoma.

**Formal analysis:** Gudani Mukoma.

**Funding acquisition:** Shane A. Norris.

**Investigation:** Gudani Mukoma.

**Methodology:** Gudani Mukoma, Edna N. Bosire, Sonja Klingberg, Shane A. Norris.

**Project administration:** Gudani Mukoma.

**Resources:** Shane A. Norris.

**Software:** Edna N. Bosire, Shane A. Norris.

**Supervision:** Shane A. Norris.

**Writing – original draft:** Gudani Mukoma.

**Writing – review & editing:** Edna N. Bosire, Sonja Klingberg, Shane A. Norris.

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
