## [Decision Letter · Decision Letter 0]

13 Mar 2023

PGPH-D-22-01947

Healthy eating and physical activity: analysing Soweto's young adults' perspectives with an intersectionality lens

Dear Dr. Mukoma,

Thank you for submitting your manuscript to PLOS Global Public Health. After careful consideration, we feel that it has merit but does not fully meet PLOS Global Public Health’s publication criteria as it currently stands. Therefore, we invite you to submit a revised version of the manuscript that addresses the points raised during the review process.

We look forward to receiving your revised manuscript.

Kind regards,

Leonor Guariguata, MPH, PhD

Academic Editor

Journal Requirements:

2. Please send a completed 'Competing Interests' statement, including any COIs declared by your co-authors. If you have no competing interests to declare, please state "The authors have declared that no competing interests exist". Otherwise please declare all competing interests beginning with the statement "I have read the journal's policy and the authors of this manuscript have the following competing interests:"

3. Please amend your detailed Financial Disclosure statement. This is published with the article. It must therefore be completed in full sentences and contain the exact wording you wish to be published.

4. Please provide separate figure files in .tif or .eps format only and remove any figures embedded in your manuscript file. Please also ensure that all files are under our size limit of 10MB.

5. We have noticed that you have cited Tables 1 and 2 in the manuscript file but there are no corresponding tables in the manuscript. Please amend your manuscript to include this table, noting that tables should not be uploaded as individual files.

6. We have noticed that you have uploaded Supporting Information files, but you have not included a list of legends. Please add a full list of legends for your Supporting Information files after the references list. 

7. In the online submission form, you indicated that "The dataset used and/or analysed during the current study are available from the corresponding author on reasonable request". All PLOS journals now require all data underlying the findings described in their manuscript to be freely available to other researchers, either 1. In a public repository, 2. Within the manuscript itself, or 3. Uploaded as supplementary information.

Additional Editor Comments (if provided):

Reviewers' comments:

Reviewer's Responses to Questions

**Comments to the Author**

1. Does this manuscript meet PLOS Global Public Health’s publication criteria? Is the manuscript technically sound, and do the data support the conclusions? The manuscript must describe methodologically and ethically rigorous research with conclusions that are appropriately drawn based on the data presented.

Reviewer #1: Yes

Reviewer #2: Yes

2. Has the statistical analysis been performed appropriately and rigorously?

Reviewer #1: I don't know

Reviewer #2: N/A

3. Have the authors made all data underlying the findings in their manuscript fully available (please refer to the Data Availability Statement at the start of the manuscript PDF file)?

Reviewer #1: Yes

Reviewer #2: Yes

4. Is the manuscript presented in an intelligible fashion and written in standard English?

Reviewer #1: Yes

Reviewer #2: Yes

5. Review Comments to the Author

Reviewer #1: the full Review has been uploaded.

Thank you for the opportunity to review this work.

Summary

This manuscript is technically sound, describing an appropriate methodological approach given the intention to explore a highly contextualized topic. There is appropriate use of the literature in the introduction and discussion.

The Authors should provide more detail on the following:

Table 1- the authors should explicitly how the descriptive statistics are displayed: mean and standard deviation, and n and percentages. This could be in a footnote.

Methods

• Page 5: Study Setting and Design:

o The setting (Lines 95-104) is well-defined but the design, from line 106, the paper should speak to the paradigm e.g. interpretivism, and also indicate what theory, theory, framework or literature informed the interview guide

o The “Participants and recruitment” section should be expanded to include the inclusion/exclusion criteria and whether other factors besides household income were used to achieve maximum variation within the sample e.g., ethnicity, religion

• Page 8: “Theoretical framework”

o Line 162: the author should indicate if Figure 1 is original or of it is adapted or adopted from another source

o “Data collection”: Line 168: please indicate the online modality/platform used for the remote interviewing

o Line 170: Give the value of the voucher in an international currency e.g., USD$ and indicate how this compared to average weekly income for example to enable the reader to see the extent to which this was an appropriate incentive

o Line 174: Please give the rationale for the gender separation to enhance the reader's understanding of the context/study design

o Lines 177- 178: Please clarify the apparent conflict between the statement “Audio recordings were transcribed verbatim and, when necessary, translated into English” and the statement in Line 166; "The FGDs were conducted in English…”

Reviewer #2: Review for ‘Healthy Eating and Physical Activity: analysing Soweto’s young adults’ perspectives with an intersectionality lens.

Overall, this is a really interesting study, on an important topic. It is well written and interesting to read. Some comments for consideration below:

Abstract:

- Line 33. You could add in how many men and how many women in addition to the total sample size.

- Line 35. It would be helpful to have something more specific regarding ‘such behaviours’ e.g, ‘barriers to engaging healthy diet and physical activity behaviours’

- Line 38. I think these results could be expanded a little here/ developed further in relation to interpretation. What does it mean when young men are describing themselves as lazy?

- Results: Are these examples of barriers or the only barriers that were described?

Introduction:

- Some explanation of how urbanisation, globalisation and commercial health determinants are indirectly affecting health, particularly in LMICs would be helpful to include here.

- Sentence on line 54-55 needs rephrasing to improve clarity.

- Could you add a line at the end is needed for why you have chosen to use an intersectional lens to explore these barriers? I know you discuss this more in the methods so something brief is fine.

Methods:

- Line 106: You could state the study design somewhere at the start of the method.

- It is not at all clear what was the purpose of collecting anthropometric data (e.g., weighing participants)- rationale for this isn’t provided but it is important. What was the use of these measures being taken for a qualitative study? If you needed this information, could you not have asked them for their BMI? There also isn’t any ethical considerations about this- how did you ensure that participants were comfortable with being weighed, for example.

- Participants: Did you purposively sample? You state that you recruited from both low and high-income households- so did you aim for homogenous or heterogenous FGDs in relation to this?

- 135: ‘how health-related behaviours turn out’ needs to be rephrased to improve clarity.

- 138-39: This sentence needs rewording for clarity. In addition, be sure to use qualitative terms like ‘explore’ rather than assess which is more quantitative.

- Line 167- a comma is needed after qualitative researchers

- What questions did you ask in the FGDs?

Results:

- Overall, the result section is very long. I think quite a bit of this can be focussed down and collapsed to reduce this section.

- Numbers of men and women needed at the beginning (or a table could provide participant details perhaps)

- I can’t see anywhere where you use the anthropometric data. So what was the purpose of collecting it?

- Domain 1- it seems to be as though your participants had a good understanding rather than basic of health in relation to sleep, check-ups, diet etc to maintain wellbeing.

- Line 263- this sub theme ‘inability to apply nutrition knowledge’ is a little bit confusing. On one hand you say that your participants have the knowledge they need regarding nutrition. However, you then go on to suggest nutrition education. It seems as though they know what to do, but there are other barriers that are getting in the way.

- Some of the theme names could be developed a bit further e.g., ‘priority is for everyone to eat’. An example would be developing this a bit further to incorporate something about financial constraints resulting in priority being to feed all family members as cheaply as possible.

- The theme ‘eating healthy means sickness’ is very interesting! Were there any other social influences outlined? Individuals in this age group are often heavily influenced by peers/ social norms.

Discussion:

- The diagram is really nice on page 19. I think this could potentially go at the start of your findings section where you first introduce your themes?

- Motivation (or lack of) was a barrier suggested by your participants. There is a lot of literature on motivation to engage in health behaviours so this could be referenced in your discussion section perhaps to address this individual factor.

6. PLOS authors have the option to publish the peer review history of their article (what does this mean?). If published, this will include your full peer review and any attached files.

**Do you want your identity to be public for this peer review?** For information about this choice, including consent withdrawal, please see our Privacy Policy.

Reviewer #1: No

Reviewer #2: No

---

## [Decision Letter · Decision Letter 1]

21 Jun 2023

Healthy eating and physical activity: analysing Soweto's young adults' perspectives with an intersectionality lens

PGPH-D-22-01947R1

Dear mr Mukoma,

We are pleased to inform you that your manuscript 'Healthy eating and physical activity: analysing Soweto's young adults' perspectives with an intersectionality lens' has been provisionally accepted for publication in PLOS Global Public Health.

Best regards,

Leonor Guariguata, MPH, PhD

Academic Editor

Reviewer Comments (if any, and for reference):

Reviewer's Responses to Questions

**Comments to the Author**

1. If the authors have adequately addressed your comments raised in a previous round of review and you feel that this manuscript is now acceptable for publication, you may indicate that here to bypass the “Comments to the Author” section, enter your conflict of interest statement in the “Confidential to Editor” section, and submit your "Accept" recommendation.

Reviewer #1: All comments have been addressed

Reviewer #2: All comments have been addressed

2. Does this manuscript meet PLOS Global Public Health’s publication criteria? Is the manuscript technically sound, and do the data support the conclusions? The manuscript must describe methodologically and ethically rigorous research with conclusions that are appropriately drawn based on the data presented.

Reviewer #1: (No Response)

Reviewer #2: Yes

3. Has the statistical analysis been performed appropriately and rigorously?

Reviewer #1: (No Response)

Reviewer #2: Yes

4. Have the authors made all data underlying the findings in their manuscript fully available (please refer to the Data Availability Statement at the start of the manuscript PDF file)?

Reviewer #1: (No Response)

Reviewer #2: Yes

5. Is the manuscript presented in an intelligible fashion and written in standard English?

Reviewer #1: (No Response)

Reviewer #2: Yes

6. Review Comments to the Author

Reviewer #1: (No Response)

Reviewer #2: Thank you for your response to the comments on this paper. You have addressed all of the points raised.

7. PLOS authors have the option to publish the peer review history of their article (what does this mean?). If published, this will include your full peer review and any attached files.

**Do you want your identity to be public for this peer review?** For information about this choice, including consent withdrawal, please see our Privacy Policy.

Reviewer #1: No

Reviewer #2: No
